# Decision to use denture adhesive in complete denture wearers after one-month run-in period: A quasi-experimental study

**Nareudee Limpuangthip[1], Wacharasak Tumrasvin[1]\*, Budsara Thongyoi[2]**

1 Department of Prosthodontics, Faculty of Dentistry, Chulalongkorn University, Bangkok, Thailand,
2 Sikhoraphum Hospital, Surin, Thailand

☯ These authors contributed equally to this work.
\* wacharasak.t@chula.ac.th

**Data Availability Statement:** All relevant data are within the article and the Supporting information files.

## Abstract

### Objectives

The aim of this study was to assess two patient-based outcomes of complete denture (CD) wearers who continued or discontinued using denture adhesive (DA) after one-month run-in period of DA use.

### Methods

This quasi-experimental study comprised 76 CD wearers. The two patient-based outcomes were oral health-related quality of life and masticatory performance, determined by the Thai-version of oral impacts on daily performances index and multiple sieve method of 20-stroke peanut mastication. Denture retention and stability were evaluated using the CU-modified Kapur criteria to classify the CD into acceptable or unacceptable quality. The outcomes were collected at 3 time points: 1) at baseline (T0), 2) after a 1-month run-in period of DA use (T1), at which time the participants decided whether to continue using DA, and 3) 1-month after continuing or discontinuing using DA (T2). Changes in the percentages of having an oral impact from T0 to T2 were evaluated using the McNemar's test. The effect of denture quality and the decision to use DA on peanut particle size across time points were assessed using repeated measures ANOVA. The peanut particle size changes in each group between time points were evaluated using the one-way repeated measures ANOVA and Tukey post-hoc comparison test.

### Results

The participants who continued using DA at T2 had greater oral impact reduction after the 1-month run-in period of DA use, whereas cleaning and emotional impacts emerged in those who discontinued using DA. At T1 and T2, the peanut particle size of the participants who continued and discontinued using DA was not significantly changed from T0, except for the acceptable CD wearers who discontinued using DA at T2, whose peanut particle size decreased from T0 to T2.

**Funding:** The present research was funded by the Faculty Research Grant, Faculty of Dentistry, Chulalongkorn University [grant number DRF 63002]. The funders had no role in study design, data collection and analysis, decision to publish, or preparation of the manuscript.

**Competing interests:** The authors declare that there is no potential conflict of interest in this study.

## Conclusions

Baseline oral impacts and their change influenced CD wearers' decision whether to continue using DA. However, masticatory performance did not affect the patients' decision.

## Introduction

People with edentulism are prevalent across countries and are predominantly older people. Many of them require complete denture (CD) treatment to improve their masticatory ability and quality of life [1]. A proportion of CD wearers report problems after denture use, mostly due to poor denture retention and stability [2–4]. To solve these problems, some CD wearers use a denture adhesive (DA) to improve the denture fit, comfort, confidence, and masticatory ability [2, 5]. However, other CD wearers do not need to use DA because they can manage their denture well, and DA use does not significantly improve the denture fit or their masticatory ability [2, 5].

The efficiency of DA use has been identified using patient-based outcomes. They can be determined subjectively through patients' perception, and objectively through professional evaluation and quantitative measures of masticatory ability. A recent systematic review revealed that DA improves denture retention and masticatory performance of CD wearers [6]. Several studies reported that, after DA use, CD wearers were more satisfied with their denture function [7–9], or reported better oral health-related quality of life (OHRQoL) [10], which reflects patients' ability to perform physical, psychological, and social activities. However, conflicting results have been reported. Some authors reported that DA use did not always enhance CD wearers' masticatory performance [11], general satisfaction, or the overall OHRQoL [12]. Typically, however, the investigators provided DA to the patients without providing a run-in period of DA use to allow them to decide whether to continue using DA [7–9, 11, 12].

According to the surveys of DA use in edentulous populations, CD wearers use DA more frequently compared with removable partial denture wearers [13]. Some denture wearers have never tried using DA, some had tried, but discontinued its use, and the others currently use a DA [2, 14]. The denture wearers who had never tried using DA was because they did not know that DAs are available. The first experience of DA use by most denture wearers was due to their dentist's recommendation, and the patient's knowledge of DA depended on information given by a dentist [2, 14]. Dentists recommend DA use, to both new and existing denture wearers, based on their experience and belief that DA provides physical and psychosocial benefits for their patients [15, 16].

Because DA experience could be an important factor in CD wearers deciding whether to use DA [2], the present study provided a run-in period of DA use for all CD wearers prior to allowing them to choose whether to continue using DA. The objective of this study was to assess the OHRQoL and masticatory performance of CD wearers before and after a 1-month run-in period of DA use.

## Materials and methods

The present study had a quasi-experimental design. The study was initially registered with Thaiclinicaltrials.org (Thai Clinical Trials Registry: TCTR20190810001), and after participant enrollment, it was registered with Clnicaltrials.gov (Trial registration ID: NCT04942262). The study protocol was approved by the Human Research Ethics Committee of the Faculty of Dentistry, Chulalongkorn University (HREC-DCU 2019–068), and was conducted in accordance

with the Declaration of Helsinki. The participants agreed to and provided written informed consent prior to study participation. The manuscript was prepared following the CONSORT guideline extension for pragmatic trials [17]. The authors confirm that all ongoing and related trials for this intervention were registered.

## Participants

The eligible participants were CD wearers recruited from patients who visited the Prosthodontic clinic at the Faculty of Dentistry, Chulalongkorn University for maintenance recall from January to December 2018. The inclusion criteria were patients who had worn removable maxillary and mandibular CDs for at least 1 year and were willing to improve their denture fit by trying DA use. The patients who wore a metal-based or implant-retained overdenture, had a history of DA use, had difficulty in responding to the interview or performing the mastication test due to physical and psychological impairment, or were unwilling to follow the study protocol were excluded from the study.

A pilot study was performed to assess the reproducibility of the study protocol and to determine the sample size of the study. The sample size was estimated using the GPower v.3.1 program based on the Z-test family and the statistical test of two independent proportions. It was found that the percentages of participants with overall oral impact reduction after 1-month of DA use between those who decided to continue and discontinue using DA were 80% ($n_1$ = 14) and 48% ($n_2$ = 11), respectively. Including a 10% drop-out rate, 38 subjects per group were required to achieve 80% power at the 5% significance level. Therefore, 76 participants were enrolled the study.

At baseline (T0), the background characteristics of the participants were collected, including general sociodemographic characteristics, CD age, and CD experiences. Oral and radiographic examinations were performed to determine the severity of the edentulous condition according to the American College of Prosthodontics (ACP) classification [18]. CD retention and stability were assessed by the same investigator who is an experienced prosthodontist (W. T.) following the CU-modified Kapur criteria [3]. The participants were classified into CD wearers with acceptable or unacceptable denture quality. The intra-examiner reliability of CD retention and stability was determined by re-evaluating 15 CD wearers 1 month later, giving a Kappa score of 0.95.

## Outcomes

The same investigator was responsible for outcome determination (B.T). The primary outcome was OHRQoL determined using the Thai-version of Oral Impacts on Daily Performances (OIDP) index [19, 20]. The participants were interviewed about whether they had difficulties in performing the following 8 daily activities: eating, speaking and pronouncing, cleaning denture and mouth, sleeping, smiling, emotional stability, social contact, and carrying out work or housework. The frequency and severity of each impact were rated using a five-point Likert scale, and their multiplication gave a condition-specific (CS) impact score. The overall oral impact score was the sum of the 8 activity scores. The prevalence of an oral impact was categorized into absence (score = 0) or presence of an oral impact (score > 0).

The secondary outcome was masticatory performance assessed using the multiple sieve method of peanut mastication [3, 21]. The participants masticated 3 g of roasted peanuts for 20 strokes in triplicate with a 10-min resting interval between each test. The comminuted peanut particles were dried and sieved through 12 standard test sieves (Test sieve; Retcsh Technology GmbH) on a vibrating sieve shaker. A simple linear regression was plotted between the cumulative weight and diameter of each sieve test. The median peanut particle size (mm) was

the sieve diameter through which 50% of the comminuted peanut particles passed. A smaller peanut particle size reflected a higher masticatory performance.

### Intervention

At baseline (T0), the participants underwent a daily 1-month run-in period of DA use. They were provided a cream-type DA (Polident®, GlaxoSmithKline, Ireland), and instructed to apply DA onto the tissue surface of their maxillary and mandibular dentures using a spot method [22]. The participants had to remove the DA and clean the denture every day after the last meal by soaking and brushing the CD with liquid soap and a soft toothbrush under running tap water [23]. Two gauze pads were used to remove the DA from the denture and oral mucosa. The participants were provided a daily calendar to make a mark after DA application each day. One dentist, who was not involved in the study, also monitored the participants' compliance by a weekly telephone notification.

The OHRQoL, masticatory performance, and CD quality were evaluated at 3 time points by the same investigator (B.T.):

1. T0, at baseline before using the DA. The baseline information regarding the outcomes were evaluated. The participants then underwent a 1-month run-in period of daily DA use,

2. T1 (T0 + 1 month), after the 1-month run-in period of DA use. The outcomes were evaluated with all participants using DA. The external prosthodontist asked the participants to choose whether they wanted to continue or discontinue using DA, and

3. T2 (T0 + 2 months), at 1-month after continuing or discontinuing DA use. At this time, some participants used DA during the outcome evaluations, while others did not, depending on the patient's decision on DA use.

Any complications caused by DA use were recorded. After the study ended, DA use by the participants with unacceptable denture quality was prohibited, and all of them had a new denture fabricated by dental students at the faculty.

### Statistical analysis

The data were analyzed using STATA version 13.0 (STATA, Chicago, IL), and the statistical significance level was set at 5%. The normality test was performed using the Shapiro Wilk test, and parametric tests were employed. The baseline characteristics of the participants who continued and discontinued using DA at T2 were described. The associations between the decision to use DA and the categorical variables were determined using the chi-square test, whereas the mean differences between two groups were determined using the independent t-test. The changes in the percentages of having an oral impact in each group between time points were evaluated using the McNemar's test. The effect of denture quality and the decision to use DA on peanut particle size across time points were assessed using two-way repeated measures ANOVA, and an interaction effect of denture quality and time was tested. The changes in the peanut particle size in each group between time points were evaluated using the one-way repeated measures ANOVA and Tukey post-hoc comparison test. The associations between peanut particle size and eating impact score at each time point were assessed using Spearman's rank correlation.

### Results

The flow diagram of the present study is illustrated in Fig 1. This study had a 100% retention rate. Therefore, the data analysis comprised 76 participants. Approximately 47% of the

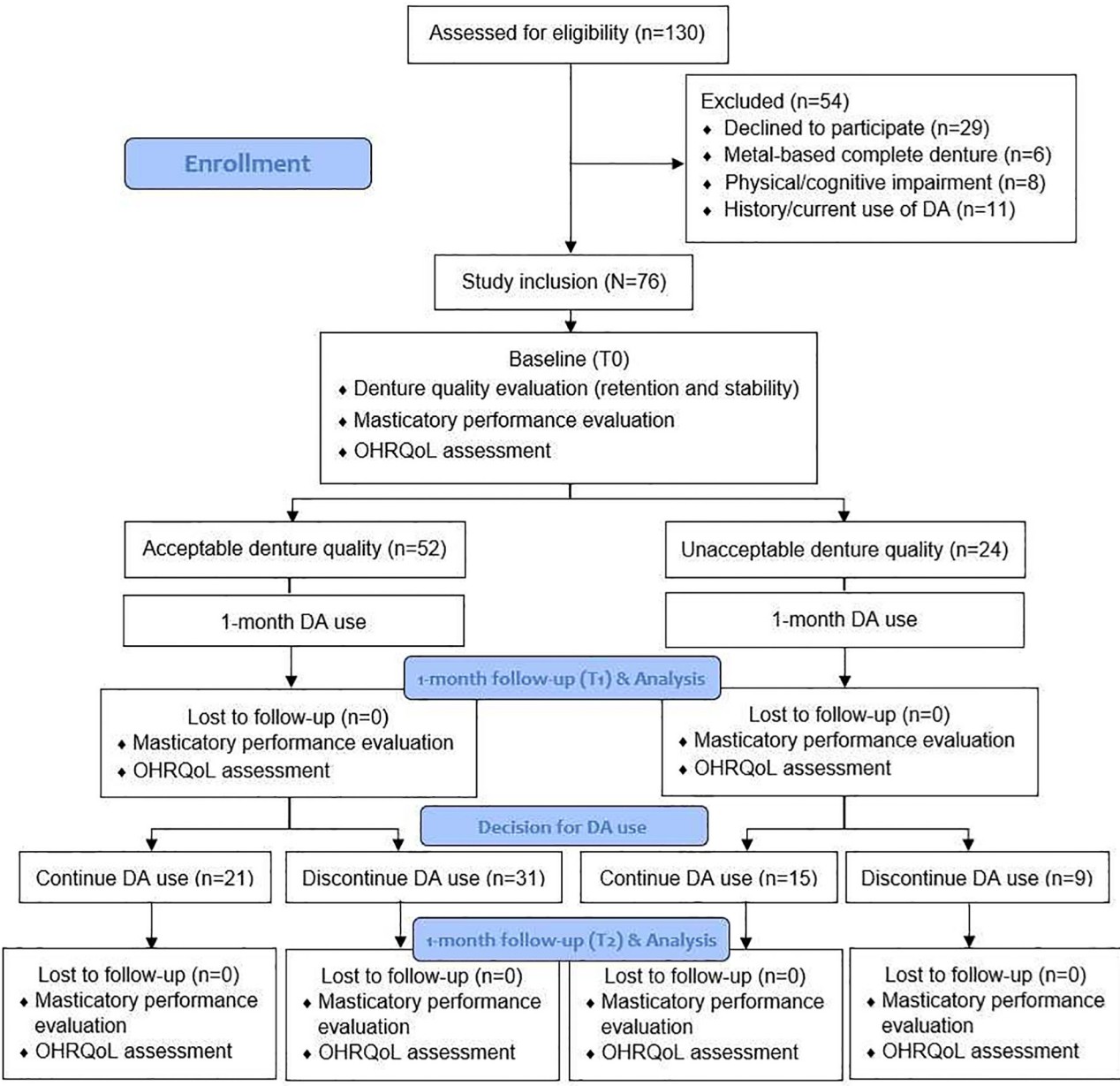

**Fig 1. Flow diagram of the study (adopted from CONSORT diagram).**

participants decided to continue using DA after the 1-month run-in period of DA use. The decision to continue using DA was not significantly associated with the participants' sociodemographic characteristics, CD age, CD experience, or ACP classification. However, it was significantly associated with the presence or absence of a baseline oral impact (Table 1). Complications were reported by two participants who discontinued using DA were due to feeling nausea and vomiting.

The participants who continued using DA at T2 more frequently presented an oral impact at T0 compared with those who discontinued using DA, regardless of CD quality (Table 2). At T0, a major problem was having an eating impact, while a cleaning impact was absent. Eating, speaking, and emotional impacts at T1 significantly decreased in the participants who

**Table 1. Baseline characteristics of the participants at T0 (N = 76).**

| Variables | | Total (%) | Decision to use DA at T2 (%) | | p-value[†] |
|---|---|---|---|---|---|
| | | | Continue using | Discontinue using | |
| | | | (n = 36) | (n = 40) | |
| Age (years): | mean ±SD | 68.0 ±8.0 | 69.4 ±7.7 | 68.2 ±7.2 | .51 |
| Sex: | male | 56.6 | 66.8 | 47.5 | .10 |
| | female | 43.4 | 33.3 | 52.5 | |
| Education: | none to primary | 60.5 | 58.7 | 67.5 | |
| | up to secondary | 30.3 | 47.8 | 27.5 | .28 |
| | up to tertiary | 9.2 | 28.6 | 5.0 | |
| Working status: | employed | 26.3 | 25.0 | 27.5 | .80 |
| | unemployed | 73.7 | 75.0 | 72.5 | |
| Having CD experience: | yes | 46.0 | 44.4 | 47.5 | .79 |
| | no | 54.0 | 55.6 | 52.5 | |
| CD years of usage: | mean ±SD | 2.6 ±1.8 | 2.3 ±1.7 | 3.0 ±2.2 | .14 |
| ACP classification: | class I | 44.8 | 47.2 | 42.5 | |
| | class II | 28.9 | 22.2 | 35.0 | .52 |
| | class III | 17.1 | 22.2 | 12.5 | |
| | class IV | 9.2 | 8.3 | 10.0 | |
| Denture quality: | acceptable | 68.4 | 58.3 | 77.5 | .07 |
| | unacceptable | 31.6 | 41.7 | 22.5 | |
| Peanut particle size (mm): | mean ±SD | 2.2 ±1.1 | 2.4 ±1.2 | 2.0 ±1.0 | .11 |
| Oral impact at T0: | absence | 51.3 | 22.2 | 77.5 | < .001 |
| | presence | 48.7 | 77.8 | 22.5 | |

[†]Analyzed using Chi-squared test or independent t-test.

continued using DA at T2. In contrast, emotional and cleaning impacts at T1 significantly emerged in the participants who discontinued using DA. The prevalence of an oral impact at T2 in the acceptable CD wearers who discontinued DA use was significantly decreased from baseline.

Peanut particle size was evaluated at each time point; a smaller peanut particle size indicated higher masticatory performance (Table 3). Among the participants with an oral impact, the peanut particle size at T0 tended to be larger in CD wearers who continued using DA, compared with that of the participants who discontinued using DA. A significant reduction in peanut particle size from T0 to T2 was present only in the acceptable CD wearers who discontinued using DA. A weak positive association between larger peanut particle size and a higher eating impact score was found at T0 (r = 0.25) and T2 (r = 0.20), however, this association was not found at T1 (r = -0.04).

## Discussion

The present study provided a 1-month run-in period of DA use for CD wearers and allowed the patients to decide whether to continue using DA. The results demonstrated that the CD wearers who continued using DA at T2 had a greater oral impact reduction after the 1-month run-in period of DA use, compared with the patients who discontinued using DA. In contrast, the masticatory performance change after 1-month of DA use between the CD wearers who continued and discontinued using DA was not significantly different. To mimic a pragmatic trial of real-world clinical use, DA was provided for both the acceptable and unacceptable CD

**Table 2. Frequency (percentage) of the participants who had oral impact at each time point.**

| Oral impact | Oral impact at T0 | Presence (N = 37) | | | | Absence (N = 39) | | | |
|---|---|---|---|---|---|---|---|---|---|
| | CD quality | Acceptable (n = 23) | | Unacceptable (n = 14) | | Acceptable (n = 29) | | Unacceptable (n = 10) | |
| | DA use at T2 | Continue (n = 15) | Discontinue (n = 8) | Continue (n = 13) | Discontinue (n = 1)† | Continue (n = 6) | Discontinue (n = 23) | Continue (n = 2)† | Discontinue (n = 8) |
| **Overall** | T0 | 15 (100)[a] | 8 (100)[a] | 13 (100)[a] | 1 (100) | 0 (0) | 0 (0)[b] | 0 (0) | 0 (0) |
| | T1 | 8 (53.3)[b] | 5 (62.5)[a] | 2 (15.4)[b] | 1 (100) | 0 (0) | 12 (52.2)[a] | 1 (50) | 3 (37.5) |
| | T2 | 3 (20)[b] | 1 (12.5)[b] | 4 (30.8)[b] | 1 (100) | 0 (0) | 0 (0)[b] | 0 (0) | 1 (12.5) |
| **Physiological** | | | | | | | | | |
| **Eat** | T0 | 14 (93.3)[a] | 7 (87.5)[a] | 13 (100)[a] | 1 (100) | 0 (0) | 0 (0) | 0 (0) | 0 (0) |
| | T1 | 4 (26.7)[b] | 7 (37.5)[b] | 2 (15.4)[b] | 1 (100) | 0 (0) | 2 (13) | 0 (0) | 0 (0) |
| | T2 | 1 (6.7)[b] | 1 (12.5)[b] | 2 (15.4)[b] | 1 (100) | 0 (0) | 0 (0) | 0 (0) | 0 (0) |
| **Speak** | T0 | 5 (33.3)[a] | 0 (0) | 7 (53.9)[a] | 1 (100) | 0 (0) | 0 (0) | 0 (0) | 0 (0) |
| | T1 | 1 (6.7)[b] | 1 (12.5) | 0 (0)[b] | 1 (100) | 0 (0) | 0 (0) | 0 (0) | 1 (12.5) |
| | T2 | 1 (6.7)[b] | 0 (0) | 0 (0)[b] | 1 (100) | 0 (0) | 0 (0) | 0 (0) | 1 (12.5) |
| **Clean** | T0 | 0 (0)[b] | 0 (0)[b] | 0 (0) | 0 (0) | 0 (0) | 0 (0)[b] | 0 (0) | 0 (0) |
| | T1 | 4 (26.7)[a] | 4 (50)[a] | 0 (0) | 1 (100) | 0 (0) | 10 (43.5)[a] | 1 (50) | 3 (37.5) |
| | T2 | 2 (20)[a] | 0 (0)[b] | 1 (7.7) | 0 (0) | 0 (0) | 0 (0)[b] | 0 (0) | 0 (0) |
| **Psychological** | | | | | | | | | |
| **Emotion** | T0 | 1 (6.7) | 0 (0)[b] | 3 (23.1) | 1 (100) | 0 (0) | 0 (0)[b] | 0 (0) | 0 (0) |
| | T1 | 2 (13.3) | 2 (25)[a] | 0 (0) | 1 (100) | 0 (0) | 4 (17.4)[a] | 0 (0) | 0 (0) |
| | T2 | 2 (13.3) | 0 (0)[b] | 1 (7.7) | 1 (100) | 0 (0) | 0 (0)[b] | 0 (0) | 0 (0) |
| **Smile** | T0 | 2 (13.3)[a] | 1 (12.5) | 2 (15.4) | 0 (0) | 0 (0) | 0 (0) | 0 (0) | 0 (0) |
| | T1 | 0 (0)[b] | 0 (0) | 0 (0) | 0 (0) | 0 (0) | 0 (0) | 0 (0) | 0 (0) |
| | T2 | 0 (0)[b] | 0 (0) | 0 (0) | 0 (0) | 0 (0) | 0 (0) | 0 (0) | 0 (0) |
| **Sleep** | T0 | 1 (6.7) | 0 (0) | 0 (0) | 0 (0) | 0 (0) | 0 (0) | 0 (0) | 0 (0) |
| | T1 | 1 (6.7) | 12.5 | 0 (0) | 0 (0) | 0 (0) | 1 (4.4) | 0 (0) | 0 (0) |
| | T2 | 1 (6.7) | 0 (0) | 0 (0) | 0 (0) | 0 (0) | 0 (0) | 0 (0) | 0 (0) |
| **Social** | | | | | | | | | |
| **Contact** | T0 | 0 (0) | 1 (12.5) | 0 (0) | 0 (0) | 0 (0) | 0 (0) | 0 (0) | 0 (0) |
| | T1 | 0 (0) | 0 (0) | 0 (0) | 0 (0) | 0 (0) | 0 (0) | 0 (0) | 0 (0) |
| | T2 | 0 (0) | 0 (0) | 0 (0) | 0 (0) | 0 (0) | 0 (0) | 0 (0) | 0 (0) |
| **Work** | T0 | 0 (0) | 1 (12.5) | 0 (0) | 0 (0) | 0 (0) | 0 (0) | 0 (0) | 0 (0) |
| | T1 | 0 (0) | 0 (0) | 0 (0) | 0 (0) | 0 (0) | 0 (0) | 0 (0) | 0 (0) |
| | T2 | 0 (0) | 0 (0) | 0 (0) | 0 (0) | 0 (0) | 0 (0) | 0 (0) | 0 (0) |

Different superscript letters indicated a significant difference between DA use within the same time point.

†No statistical analysis between time points due to a very low number of sample (n = 1, 2).

wearers, however, its use was limited to 1–2 months due to the duration of our study. After the study ended, DA use in unacceptable CD wearers was prohibited and all of them had a new CD fabricated.

The present study used the OIDP to evaluate the participants' OHRQoL. Previous studies evaluated the effect of DA use on the OHRQoL of CD wearers using the Oral Health Impacts Profile for Edentulous Patients (OHIP-EDENT) [12], short-formed Oral Health Impacts Profile (OHIP-14) [24], or Geriatric Oral Health Assessment Index (GOHAI) [25]. However, the OHIP and GOHAI evaluation do not include cleaning the oral mucosa and denture, which is a major problem caused by DA use. Thus, an additional question had to be included to detect a

**Table 3. Peanut particle size (mm; mean ±SD) at each time point.**

| Oral impact at T0 | | Presence (N = 37) | | | | Absence (N = 39) | | | |
|---|---|---|---|---|---|---|---|---|---|
| CD quality | | Acceptable (n = 23) | | Unacceptable (n = 14) | | Acceptable (n = 29) | | Unacceptable (n = 10) | |
| DA use at T2 | | Continue (n = 15) | Discontinue (n = 8) | Continue (n = 13) | Discontinue† (n = 1) | Continue (n = 6) | Discontinue (n = 23) | Continue† (n = 2) | Discontinue (n = 8) |
| Time: | T0 | 2.37 ±1.26 | 2.08 ±0.92[a] | 2.93 ±1.28 | 1.53 (N/A) | 1.78 ±0.78 | 1.90 ±1.12 | 1.70 ±0.95 | 2.37 ±0.97 |
| | T1 | 2.04 ±0.82 | 1.87 ±0.67[a] | 2.46 ±0.83 | 1.14 (N/A) | 1.42 ±0.48 | 1.71 ±0.93 | 2.58 ±0.48 | 1.85 ±1.44 |
| | T2 | 2.35 ±0.91 | 1.52 ±0.61[b] | 2.46 ±0.92 | 1.32 (N/A) | 1.43 ±0.31 | 1.43 ±0.31 | 2.11 ±0.92 | 2.23 ±1.83 |

N/A, not applicable. Different lower case letters indicate significant difference between time points within the same column (p<0.05).

†No statistical analysis between the decision of DA use and time points due to a very low number of samples in the subgroup (n = 1, 2).

cleaning problem [7]. Because the OIDP covers a cleaning aspect, our results indicate using the OIDP as a subjective indicator for evaluating an impact of DA use on the OHRQoL of CD wearers.

Our findings revealed that presence of an oral impact and its changes after the 1-month run-in period of DA use influenced the decision to continue using DA in CD wearers. In CD wearers who decided to continue DA use, eating and speaking impacts were generally decreased, while a cleaning impact significantly emerged after the 1-month of DA use. Despite decreasing eating and speaking impacts, the decision to discontinue using DA was commonly due to an emerging cleaning impact. At baseline, an emotional disturbance was caused by an eating impact; however, at T1, it was caused by a cleaning impact. As found in previous studies in CD wearers [2, 7, 14], common reasons for stopping using DA are its difficulty to remove from the denture and oral mucosa, and an unpleasant taste and texture. Therefore, a patient's decision to use DA may depend on an emerging oral impact caused by DA that is more troublesome compared with the problems solved by DA use. Due to this cleaning problem, DA use may not be suitable for CD wearers who have poor manual dexterity or denture hygiene practice.

The masticatory performance of CD wearers and its change after 1-month of DA use did not significantly influence the decision to continue using DA. Previous studies reported that DA use in newly-fabricated CD wearers significantly improved their masticatory performance [6, 26–28], both in normal and resorbed ridges [27]. However, our study found no masticatory performance improvement after 1-month of DA use, either in well-fitting or ill-fitting denture, or in any patients' condition based on ACP classification. A significant association between the presence of an eating impact and lower masticatory performance was observed at baseline, but not at T1 or T2. A patient-reported oral impact might be more sensitive to changes caused by DA use compared with an objective masticatory performance, which can be influenced by several factors, such as patient's age, and the occlusal contact area of the artificial teeth [4]. Also, it might be possible that the patients with ill-fitting denture had already adapted to it and learned to manipulate the denture.

Professional evaluation of CD retention and stability did not affect the patients' decision to continue using DA. However, approximately 42% of the CD wearers who continued using DA possessed unacceptable CD retention and stability, and DA use significantly reduced the oral impacts of these CD wearers. It has been established that DA should not be used in ill-fitting denture because it would compromise the adaptation between the underlying tissue and a denture, leading to denture supporting tissue traumatization without the patients' awareness. Because DA is an over-the counter product, its application under unsuitable oral or denture

condition could be harmful to the CD wearers without the patients' notice. Therefore, DA use should be prescribed by a dental professional after a denture retention and stability evaluation. In addition, dentists should provide a maintenance recall for periodic evaluation of denture quality to prevent inappropriate DA use in CD wearers.

One interesting finding was present in the acceptable CD wearers who had an oral impact at baseline but discontinued using DA at T2. In this subgroup, their OHRQoL and masticatory performance continuously improved from baseline to T1 and T2, despite discontinued DA use at T2. In contrast with a previous pilot study in well-fitting CD wearers, the patients who continued using DA for 6 months demonstrated greater OHRQoL improvement and slightly higher masticatory parameters, compared with those who discontinued using DA at 3 months [10]. Our findings imply that a 1-month run-in period of DA use may assist the acceptable CD wearers in controlling and adapting to their denture during function. In acceptable CD wearers with a persisting oral impact, other treatment options might be considered, such as an implant-retained mandibular overdenture [29].

The present study found no significant associations between the decision to continue using DA and patients' age, sex, previous CD experience, denture age, or the patient's ACP classification. Previous cross-sectional surveys in removable denture wearers demonstrated an association between DA use and patient-related factors rather than denture characteristics. They concluded that DA use was more common in smokers, people who regularly visit a dental clinic [13], had been wearing the denture for a shorter time, and had a lower frequency of denture use [30]. However, many denture wearers have never tried using DA due to a lack of DA knowledge and being unaware that DAs are available. Unless CD wearers have tried using DA, patient-related conditions and CD experience might not be the appropriate indicator to predict whether CD wearers would prefer DA use.

Based on our findings, the three considerations for identifying the CD wearers who would benefit from DA use were a professional assessment of CD retention and stability, a patient-reported baseline oral impact, and oral impact changes after a 1-month run-in period of DA use. With the presence of an oral impact, the unacceptable CD wearers would not be eligible for DA use, whereas the acceptable-quality CD wearers might undergo a 1-month run-in period of DA use. The oral impact should be then re-evaluated. The patients would decide whether to continue using DA based on the emerging and decreasing oral impacts. Despite discontinuing DA use, the temporary use of DA for 1 month may assist some well-fitting CD wearers in adapting to their denture.

The present study has some limitations. First, only a cream-type DA was provided to the patients, because it is the most available commercial form in Thailand, whereas a powder-type is limited to professional use in dental clinics. Second, the 1-month follow-up period was relatively short to determine the stable results with patient-reported outcome measures, and it remains unidentified whether the patients would continue using DA based on its additional cost. However, this was because approximately half of the patients with ill-fitting denture had a new CD fabricated within a few months. During this time, wearing an ill-fitting denture was prohibited, otherwise, a soft or hard denture lining material was applied on the tissue surface of the denture. Future studies with a longer follow-up period are recommended for better understanding the effect of DA use in CD wearers and their adaptability to DA use. A cost-benefit analysis should be conducted to evaluate all potential costs that are associated with long-term DA use.

## Conclusions

Within the limitations of this clinical study, it can be concluded that presence of an oral impact at baseline and its change influenced the CD wearers' decision whether to continue using DA.

Furthermore, masticatory performance at baseline and its change did not influence on the patients' decision to continue using DA.

## Supporting information

**S1 Checklist. CONSORT 2010 checklist information.**
(DOCX)

**S1 Dataset. Raw data of the present study.**
(XLSX)

**S1 File. Study protocol in Thai.**
(PDF)

**S2 File. Study protocol in English.**
(PDF)

## Acknowledgments

The authors gratefully acknowledge Dr.Kevin Tompkins for language revision of the manuscript.

## Author Contributions

**Conceptualization:** Nareudee Limpuangthip, Wacharasak Tumrasvin, Budsara Thongyoi.

**Data curation:** Nareudee Limpuangthip, Wacharasak Tumrasvin, Budsara Thongyoi.

**Formal analysis:** Nareudee Limpuangthip, Budsara Thongyoi.

**Funding acquisition:** Nareudee Limpuangthip.

**Investigation:** Wacharasak Tumrasvin, Budsara Thongyoi.

**Methodology:** Nareudee Limpuangthip, Wacharasak Tumrasvin, Budsara Thongyoi.

**Project administration:** Nareudee Limpuangthip, Wacharasak Tumrasvin.

**Resources:** Wacharasak Tumrasvin.

**Software:** Nareudee Limpuangthip.

**Supervision:** Nareudee Limpuangthip, Wacharasak Tumrasvin.

**Validation:** Nareudee Limpuangthip, Wacharasak Tumrasvin.

**Visualization:** Nareudee Limpuangthip, Wacharasak Tumrasvin.

**Writing – original draft:** Nareudee Limpuangthip, Wacharasak Tumrasvin, Budsara Thongyoi.

**Writing – review & editing:** Nareudee Limpuangthip, Wacharasak Tumrasvin, Budsara Thongyoi.

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
