## [Decision Letter · Decision Letter 0]

28 Apr 2022

PONE-D-22-08584Decision to Use Denture Adhesive in Complete Denture Wearers After One-Month Trial: A Quasi-Experimental StudyPLOS ONE

Dear Dr. Tumrasvin,

Thank you for submitting your manuscript to PLOS ONE. After careful consideration, we feel that it has merit but does not fully meet PLOS ONE’s publication criteria as it currently stands. Therefore, we invite you to submit a revised version of the manuscript that addresses the points raised during the review process.

We look forward to receiving your revised manuscript.

Kind regards,

Gaetano Isola, Ph.D.

Academic Editor

PLOS ONE

Journal Requirements:

2. Thank you for submitting your clinical trial to PLOS ONE and for providing the name of the registry and the registration number. The information in the registry entry suggests that your trial was registered after patient recruitment began. PLOS ONE strongly encourages authors to register all trials before recruiting the first participant in a study.

1) your reasons for your delay in registering this study (after enrolment of participants started);

2) confirmation that all related trials are registered by stating: “The authors confirm that all ongoing and related trials for this drug/intervention are registered”.

4.Thank you for stating the following in the Funding Section of your manuscript: 

"The present research was funded by the Faculty Research Grant, Faculty of Dentistry, Chulalongkorn University [grant number DRF 63002]. The funders had no role in study design, data collection and analysis, decision to publish, or preparation of the manuscript."

"The present research was funded by the Faculty Research Grant, Faculty of Dentistry, Chulalongkorn University [grant number DRF 63002]. The funders had no role in study design, data collection and analysis, decision to publish, or preparation of the manuscript."

Additional Editor Comments:

The manuscript still requests some major revisions

Reviewers' comments:

Reviewer's Responses to Questions

**Comments to the Author**

1. Is the manuscript technically sound, and do the data support the conclusions?

Reviewer #1: Yes

Reviewer #2: No

2. Has the statistical analysis been performed appropriately and rigorously? 

Reviewer #1: Yes

Reviewer #2: No

3. Have the authors made all data underlying the findings in their manuscript fully available?

Reviewer #1: Yes

Reviewer #2: Yes

4. Is the manuscript presented in an intelligible fashion and written in standard English?

Reviewer #1: Yes

Reviewer #2: Yes

5. Review Comments to the Author

Reviewer #1: Authors conducted an interesting paper, for which I present some comments/suggestions in the attached pdf file. In summary, the paper describes a relevant clinical study on oral health that may benefit from some clarification and slightly different writing. I have some concern regarding the short term follow-up (1 month), as probably the major issue I could find -- please justify and/or explain the limitations arisen by the timeline in the Discussion section

Reviewer #2: A clinical trial was conducted which aimed to assess health-related quality of life and masticatory performance at three time points. Since the statistical analyses are not comprehensive, the conclusions are unclear.

Major revision:

Comprehensive statistical methods are required to analyze the repeated measures data. Additionally, test the interaction effects.

Suggested revisions:

1- Abstract: Spell out all acronyms and abbreviations.

2- Abstract: The phrasing of the statistical methods described in the abstract is confusing. Only one of the following method should be used for each comparison: McNemar’s, chi-square or Fisher’s exact test. McNemar’s is used with dependent observations. The other two are for comparing independent observations.

3- Abstract and Statistical analysis section: Does repeated ANOVA refer to a “repeated measures ANOVA” analysis? Clarify.

4- Page 5: Replace “during January” with “from January.”

5- Page 5: Sample size and statistical power: State the statistical testing method which attains 80% power.

6- Page 8, Statistical analysis:

- Does the first sentence imply that statistical significance was set at 5%? Clarify the statement.

- Chi-square tests are used for testing associations and independent t-tests are used to compare differences in means between two groups. Clarify this statement.

- Analyze differences in peanut particle size between groups and across time points using a repeated measures ANOVA. Test the interaction effect of group by time. If, however, the peanut particle size is not normally distributed use a linear mixed model. Test the indicated interaction effect.

7- Table 1:

- In addition to the percentages, provide the corresponding frequencies.

- Consider replacing “CD age” with “CD years of usage.”

- Indicate if the distribution of the data was checked for normality prior to applying t-tests.

8- Table 2:

- In addition to the percentages, provide the corresponding frequencies.

- To model repeated measures data, use mixed effects logistic regression models instead of simple logistic regression models.

9- Page 11: The letter r is used to indicate the estimate of rho. Thus “r=0.30” and “r=0.08” would be the standard presentation. Is the correlation coefficient the same at T1 and T2? If not, present both. Note: The p-value associated with a correlation is a test of the null hypothesis: correlation equal to zero; however, the absolute magnitude of the coefficient indicates the strength of the linear relationship between two variables. In general, the strength or correlation coefficient is the more important statistic to focus on.

Below is a table for interpreting correlation coefficients:

Coefficient (absolute value) Interpretation

0.90 - 1.0 Very Strong

0.70 - 0.89 Strong

0.40 - 0.69 Moderate

0.10 - 0.39 Weak

10- To assist in the review process, add line numbers to the document.

6. PLOS authors have the option to publish the peer review history of their article (what does this mean?). If published, this will include your full peer review and any attached files.

Reviewer #1: No

Reviewer #2: No

---

## [Author Response · Author response to Decision Letter 0]

30 Jul 2022

The authors are pleased to submit our revised manuscript ID. PONE-D-22-08584. The title has been revised to ‘Decision to Use Denture Adhesive in Complete Denture Wearers After One-Month Run-In Period: A Quasi-Experimental Study’. The requested revisions have been made in the manuscript in track changes, and our point-by-point responses are below.

Reviewer #1: 

Comment: Authors conducted an interesting paper, for which I present some comments/suggestions in the attached pdf file. In summary, the paper describes a relevant clinical study on oral health that may benefit from some clarification and slightly different writing. I have some concern regarding the short-term follow-up (1 month), as probably the major issue I could find -- please justify and/or explain the limitations arisen by the timeline in the Discussion section

Response: The limitation on short follow-up duration has been added to the ‘Discussion’ section, stating that ‘the 1-month follow-up period was relatively short to determine the stable results with patient-reported outcome measures, and it remains unidentified whether the patients would continue using DA based on its additional cost. However, this was because approximately half of the patients with ill-fitting denture had a new CD fabricated within a few months. During this time, wearing an ill-fitting denture was prohibited, otherwise, a soft or hard denture lining material was applied on the tissue surface of the denture.’

Other responses according to comments in PDF file of the manuscript are as follows:

Abstract

1. Comment: Please use another word instead of ‘trial’.

Response: The word ‘trial’ of denture adhesive use has been changed to ‘run-in period’ of denture adhesive use.

Introduction

1. Comment on study objective: This part is confusing and does not put the objectives of this study in a clear manner. I suggest rewriting it, by stating PICO components of a research question in your objectives, and then mention that all participants had used DA during a 1-month run-in period. 

Response: The objective of this study has been revising by stating the PICO components. The last paragraph of Introduction section has been revised to ‘Because DA experience could be an important factor in CD wearers deciding whether to use DA, the present study provided a run-in period of DA use for all CD wearers prior to allowing them to choose whether to continue using DA. The objective of this study was to assess the OHRQoL and masticatory performance of CD wearers before and after a 1-month run-in period of DA use.’

Materials and Methods

1. Comment: Which of the CONSORT guideline being used, main or extensions? Please specify and cite the reference.

Response: The CONSORT guideline extension for pragmatic trials was used. The reference has been added in the first paragraph of the Materials and Methods section. 

2. Comment on participants subsection: Please describe the recruitment protocol. Unless you created a random number algorithm to select participants from your former patient database, the selection possibly was not random BTW (in such case, avoid the term "randomly")

Response: The word ‘randomly’ has been removed from the sentence.

3. Comment on participants subsection: Was there any restriction to the timing of being edentulous, case complexity, age, systemic status, presence of implant and other factors that may influence the performance of CDs?

Response: There was no restriction to age, the timing of being edentulous, and case complexity (ACP classification). Although these factors may influence the CDs performance, they were equally distributed between the participants who continued and discontinued using denture adhesive (as shown in Table 1). The exclusion criteria were the patients who wore a metal-based or implant-retained overdenture or had difficulty in responding to the interview or performing the mastication test due to physical and psychological impairment. 

Tables

1. Comment on Table 2. Please remove decimal from the percentages.

Response: The decimals have been removed from the percentage with an integer number to facilitate data visualization. However, decimal remains in non-integer number for accurate data. The frequency of the participant has been added.

Reviewer #2: 

A clinical trial was conducted which aimed to assess health-related quality of life and masticatory performance at three time points. Since the statistical analyses are not comprehensive, the conclusions are unclear.

Major revision:

Comprehensive statistical methods are required to analyze the repeated measures data. Additionally, test the interaction effects.

Suggested revisions:

1- Comment on Abstract: Spell out all acronyms and abbreviations.

Response: All acronyms and abbreviations in the abstract have been spelled out. 

2- Comment on Abstract: The phrasing of the statistical methods described in the abstract is confusing. Only one of the following methods should be used for each comparison: McNemar’s, chi-square or Fisher’s exact test. McNemar’s is used with dependent observations. The other two are for comparing independent observations.

Response: The sentences have been revised to ‘Changes in the percentages of having an oral impact from T0 to T2 were evaluated using the McNemar’s test. The effect of denture quality and decision to use DA on peanut particle size across time points were assessed using repeated measures ANOVA, and an interaction effect was found. The peanut particle size changes in each group between time points were evaluated using the one-way repeated measures ANOVA and Tukey post-hoc comparison test.’

3- Comment on Abstract and Statistical analysis section: Does repeated ANOVA refer to a “repeated measures ANOVA” analysis? Clarify.

Response: The word ‘repeated measures ANOVA’ has been used throughout the manuscript.

4- Comment on Page 5: Replace “during January” with “from January.”

Response: ‘during January’ has been replaced with ‘from January’. 

5- Comment on Page 5: Sample size and statistical power: State the statistical testing method which attains 80% power.

Response: The statistical testing method for attaining 80% power is described in Page 6 (line 101), stating that ‘The sample size was estimated using the GPower v.3.1 program based on the Z-test family and the statistical test of two independent proportions.

6- Comment on Page 8, Statistical analysis:

- Does the first sentence imply that statistical significance was set at 5%? Clarify the statement.

- Chi-square tests are used for testing associations and independent t-tests are used to compare differences in means between two groups. Clarify this statement.

- Analyze differences in peanut particle size between groups and across time points using a repeated measures ANOVA. Test the interaction effect of group by time. If, however, the peanut particle size is not normally distributed use a linear mixed model. Test the indicated interaction effect.

Response: 

- The first sentence has been revised to ‘Data were analyzed using STATA version 13.0 (STATA, Chicago, IL), and the statistical significance level was set at 5%.’

- The statement in line 162–165 has been revised to ‘The associations between the decision to use DA and the categorical variables were determined using chi-square test, whereas the mean differences between two groups were determined using the independent t-test.’

- Descriptions on the differences in peanut particle size between groups across time points have been revised to ‘The effect of denture quality and decision to use DA on peanut particle size across time points were assessed using a two-way repeated measures ANOVA, and an interaction effect of denture quality and time was found. The changes in the peanut particle size in each group between time points were evaluated using the one-way repeated measures ANOVA and Tukey post-hoc comparison test.’

7- Comment on Table 1:

- In addition to the percentages, provide the corresponding frequencies.

Response: The corresponding frequencies have been added in the Table 1.

- Consider replacing “CD age” with “CD years of usage.”

Response: The ‘CD age’ has been replaced with ‘CD years of usage’ in the Table 1.

- Indicate if the distribution of the data was checked for normality prior to applying t-tests.

Response: The description on normality testing has been added, stating that ‘The normality test was performed using Shapiro Wilk test, and parametric tests were employed.’

8- Comment on Table 2:

- In addition to the percentages, provide the corresponding frequencies.

Response: The corresponding frequencies have been added in the Table 2.

- To model repeated measures data, use mixed effects logistic regression models instead of simple logistic regression models.

Response: The McNemar test were used instead of regression models because the number of participants in some subgroups was too small (n=1, 2) which result in data being dropped-out when using a regression analysis.

9- Comment on Page 11: The letter r is used to indicate the estimate of rho. Thus “r=0.30” and “r=0.08” would be the standard presentation. Is the correlation coefficient the same at T1 and T2? If not, present both. Note: The p-value associated with a correlation is a test of the null hypothesis: correlation equal to zero; however, the absolute magnitude of the coefficient indicates the strength of the linear relationship between two variables. In general, the strength or correlation coefficient is the more important statistic to focus on. Below is a table for interpreting correlation coefficients:

Coefficient (absolute value) Interpretation

0.90 - 1.0 Very Strong

0.70 - 0.89 Strong

0.40 - 0.69 Moderate

0.10 - 0.39 Weak

Response: The p-value of the Spearman correlation has been removed. The coefficient interpretations have been added according to the strength of association.

10- Comment: To assist in the review process, add line numbers to the document.

Response: The line numbers have been added in the manuscript.

Sincerely yours,

Wacharasak Tumrasvin

Corresponding author

---

## [Decision Letter · Decision Letter 1]

24 Aug 2022

PONE-D-22-08584R1Decision to use denture adhesive in complete denture wearers after one-month run-in period: A quasi-experimental studyPLOS ONE

Dear Dr. Tumrasvin,

Thank you for submitting your manuscript to PLOS ONE. After careful consideration, we feel that it has merit but does not fully meet PLOS ONE’s publication criteria as it currently stands. Therefore, we invite you to submit a revised version of the manuscript that addresses the points raised during the review process.

We look forward to receiving your revised manuscript.

Kind regards,

Gaetano Isola, Ph.D.

Academic Editor

PLOS ONE

Journal Requirements:

Additional Editor Comments:

Lines 174-6: In the "Statistical analysis" section, describe only the statistical methods. Do not include the results. Remove the fact that an interaction effect was found. Simply indicate that an interaction effect was tested.

Reviewers' comments:

Reviewer's Responses to Questions

**Comments to the Author**

1. If the authors have adequately addressed your comments raised in a previous round of review and you feel that this manuscript is now acceptable for publication, you may indicate that here to bypass the “Comments to the Author” section, enter your conflict of interest statement in the “Confidential to Editor” section, and submit your "Accept" recommendation.

Reviewer #1: All comments have been addressed

Reviewer #2: (No Response)

2. Is the manuscript technically sound, and do the data support the conclusions?

Reviewer #1: Yes

Reviewer #2: Yes

3. Has the statistical analysis been performed appropriately and rigorously? 

Reviewer #1: Yes

Reviewer #2: Yes

4. Have the authors made all data underlying the findings in their manuscript fully available?

Reviewer #1: Yes

Reviewer #2: Yes

5. Is the manuscript presented in an intelligible fashion and written in standard English?

Reviewer #1: Yes

Reviewer #2: Yes

6. Review Comments to the Author

Reviewer #1: The authors made significant improvements in their manuscript. I am satisfied with the present version.

Reviewer #2: Minor revision (Note that line numbers refer to those in the tracked changes version.)

Lines 174-6: In the "Statistical analysis" section, describe only the statistical methods. Do not include the results. Remove the fact that an interaction effect was found. Simply indicate that an interaction effect was tested.

7. PLOS authors have the option to publish the peer review history of their article (what does this mean?). If published, this will include your full peer review and any attached files.

Reviewer #1: No

Reviewer #2: No

---

## [Author Response · Author response to Decision Letter 1]

17 Sep 2022

The authors are pleased to submit our revised manuscript ID. PONE-D-22-08584R1, entitled ‘Decision to Use Denture Adhesive in Complete Denture Wearers After One-Month Run-In Period: A Quasi-Experimental Study’. The authors have changed the sequence of reference lists corresponding to the previous changes at first revision. The requested revisions have been made in the manuscript in track changes, and our point-by-point responses are below.

Reviewer #1: 

Comment: The authors made significant improvements in their manuscript. I am satisfied with the present version.

Response: The authors would like to thank you for reviewers’ comments which make the substantial improvement of the manuscript.

Reviewer #2: 

Comment: Minor revision (Note that line numbers refer to those in the tracked changes version.)

Lines 174-6: In the "Statistical analysis" section, describe only the statistical methods. Do not include the results. Remove the fact that an interaction effect was found. Simply indicate that an interaction effect was tested.

Response: In the statistical analysis section, the sentence has been revised to ‘The effect of denture quality and the decision to use DA on peanut particle size across time points were assessed using two-way repeated measures ANOVA, and an interaction effect of denture quality and time was tested.’

Sincerely yours,

Wacharasak Tumrasvin

Corresponding author

---

## [Decision Letter · Decision Letter 2]

13 Oct 2022

Decision to use denture adhesive in complete denture wearers after one-month run-in period: A quasi-experimental study

PONE-D-22-08584R2

Dear Dr. Tumrasvin,

We’re pleased to inform you that your manuscript has been judged scientifically suitable for publication and will be formally accepted for publication once it meets all outstanding technical requirements.

Kind regards,

Gaetano Isola, Ph.D.

Academic Editor

PLOS ONE

Additional Editor Comments (optional):

Reviewers' comments:

Reviewer's Responses to Questions

**Comments to the Author**

1. If the authors have adequately addressed your comments raised in a previous round of review and you feel that this manuscript is now acceptable for publication, you may indicate that here to bypass the “Comments to the Author” section, enter your conflict of interest statement in the “Confidential to Editor” section, and submit your "Accept" recommendation.

Reviewer #2: All comments have been addressed

2. Is the manuscript technically sound, and do the data support the conclusions?

Reviewer #2: (No Response)

3. Has the statistical analysis been performed appropriately and rigorously? 

Reviewer #2: (No Response)

4. Have the authors made all data underlying the findings in their manuscript fully available?

Reviewer #2: (No Response)

5. Is the manuscript presented in an intelligible fashion and written in standard English?

Reviewer #2: (No Response)

6. Review Comments to the Author

Reviewer #2: (No Response)

7. PLOS authors have the option to publish the peer review history of their article (what does this mean?). If published, this will include your full peer review and any attached files.

Reviewer #2: No

---

## [Editor Report · Acceptance letter]

23 Nov 2022

PONE-D-22-08584R2 

Decision to Use Denture Adhesive in Complete Denture Wearers After One-Month Run-In Period: A Quasi-Experimental Study 

Dear Dr. Tumrasvin:

I'm pleased to inform you that your manuscript has been deemed suitable for publication in PLOS ONE. Congratulations! Your manuscript is now with our production department. 

Kind regards, 

on behalf of

Prof. Gaetano Isola 

Academic Editor

PLOS ONE